# WAV2TOK: DEEP SEQUENCE TOKENIZER FOR AUDIO RETRIEVAL

**Adhiraj Banerjee, Vipul Arora**
Department of Electrical Engineering
Indian Institute of Technology Kanpur, India
{adhiraj,vipular}@iitk.ac.in

## ABSTRACT

Search over audio sequences is a fundamental problem. In this paper, we propose a method to extract concise discrete representations for audio that can be used for efficient retrieval. Our motivation comes from orthography which represents speech of a given language in a concise and distinct discrete form. The proposed method, **wav2tok**, learns such representations for any kind of audio, speech or non-speech, from pairs of similar audio. **wav2tok** compresses the query and target sequences into shorter sequences of tokens that are faster to match. The learning method makes use of CTC loss and expectation-maximization algorithm, which are generally used for supervised automatic speech recognition and for learning discrete latent variables, respectively. Experiments show the consistent performance of **wav2tok** across two audio retrieval tasks: music search (query by humming) and speech search via audio query, outperforming state-of-the-art baselines.

## 1 INTRODUCTION

Sequence Retrieval aims at retrieving sequences similar to a query sequence, with the constraint that an ordered alignment exists between the query and the target sequence. In this paper, we address the following problem: Can we extract discrete tokens from any continuous signal for the purpose of retrieval of similar signals? This problem has deep connections with tasks such as child language acquisition, music cognition and learning languages without written forms. Some direct applications of the proposed task include speech search, where the order of constituent units, such as phonemes, syllables or words, remains same; and music search – query by humming or query by example – where the order of constituent units, such as relative notes or phrases, remains same. Apart from audio, the problem extends to tasks such as handwritten word search and gesture search.

One can define similarity metrics over sequences using methods based on Dynamic Time Warping (DTW) (Müller, 2007). These methods are inefficient if the sequences are continuous valued and have high sampling rates. Moreover, they depend on matching hand-made features, which are ineffective in the face of high variability of query sequences.

Problems such as *spoken term detection* involve detection of a query utterance in a long speech audio. The search space is huge, and performing DTW based search of query takes long time (Rodriguez-Fuentes et al., 2014). A more efficient way of sequence retrieval is by mapping them to sequences of discrete tokens. Automatic speech recognition (ASR) can be employed for this purpose (Mamou et al., 2013). However, ASR training requires knowledge of basic units of transcription. The popularly used units are phonemes and graphemes. This method thus becomes language dependent. Non-linguistic sounds, such as cough and sneeze, could be mapped to certain tokens defined for them. This approach could not be used when precise tokens are not defined, e.g., music search. ]

In *query by humming* based music search, audio is mapped to discrete melody-related tokens, such as notes, and these token sequence are matched for search (Unal et al., 2008). However, several music traditions do not have precise transcription systems. There, one can tell if two pieces, or motifs, are similar but cannot precisely transcribe them to tokens. The embellishments used in music could be too dynamic to be transcribed precisely. Moreover, when a musically untrained user sings a query, s/he cannot hit the right notes matching the target song. So the matching could rely on several factors other than notes, such as phonemes of lyrics (Mesaros & Virtanen, 2010), onset times

(rhythm) (Kosugi et al., 2000), and note transitions (Ranjan & Arora, 2020). Hence, the tokens to be used may not be derived from notes alone.

In this way, each tokenizer - for speech, music or other signals, in general - uses domain-specific hand-made tokens defined by a domain expert. In this paper, we propose a tokenizer to map audio sequences to sequences of discrete tokens with an aim of retrieval. The mapping is learned only from pairs of similar audio sequences. The tokens are not defined manually but correspond to distinct semantic units learned from pairs of similar audio sequences. The method is general and can be applied to signals other than audio. In this paper, we apply the proposed method to speech and music audio search, for the problems of spoken term detection and query by humming, respectively.

The proposed method, named **wav2tok**, encodes audio via a BiLSTM (Schuster & Paliwal, 1997) network. The encoder-generated representations are then mapped to discrete tokens via a $K$-means vector quantizer network. Each discrete token corresponds to a discrete representation in the vector quantizer's codebook which is initialized and updated via offline $K$-means clustering only.

**wav2tok** is trained with pairs of similar audio sequences in a self-supervised fashion without any transcription using a novel training algorithm. For each pair, we average the encoder-generated representations, which map to the same token, by the $K$-means vector quantizer network to generate a prototype for that token. We then perform a contrastive learning task to increase the similarity between the generated prototype for a particular token and the quantizer codebook discrete representation corresponding to the same token. We simultaneously minimize the edit distance between the token sequences generated from each sequence in the pair via Connectionist temporal classification (CTC) (Graves et al., 2006) framework to constrain both sequences to get mapped to the same token sequence.

We compare **wav2tok** to state-of-the-art (SOTA) methods for discrete representation learning, such as **wav2vec** 2.0, and SOTA ASR models fine-tuned to perform phonetic tokenization. We evaluate the generalization capability of the tokens generated by the models on search experiments, namely, query-by-humming and spoken term detection. **wav2tok** outperforms the baselines in performance and uses much lesser trainable parameters, ensuring faster inference and deployment.

## 2 RELATED WORK

**Sequence Labelling.** With expert-defined tokens, various methods are popularly used for mapping sequences to tokens. In conventional methods, Hidden Markov Models (Rabiner & Juang, 1986) and Conditional Random Fields (Lafferty et al., 2001) have been popularly used for sequence labeling. These methods involve a significant amount of domain knowledge and many assumptions to make tractable models, which are avoided by End-to-End learning models such as Recurrent Neural Networks (RNNs) using Connectionist Temporal Classification framework (Graves et al., 2006). Sequence labeling can be used for sequence retrieval by converting the sequences to tokens, which are easy to search over. But this approach inevitably depends upon expert-defined tokens.

**Unsupervised Speech Representation Learning.** Automatic Speech Recognition systems are pre-trained on large amounts of untranscribed speech data to generate SOTA continuous representations which encode the slowly varying phoneme features in raw speech. The representations are then mapped to phoneme tokens via Connectionist Temporal Classification (CTC) (Graves et al., 2006) fine-tuning on a small amount of transcribed audio. Works like Contrastive Predictive Coding (CPC) (van den Oord et al., 2018), Autoregressive Predictive Coding (APC) (Chung & Glass, 2020), and **wav2vec** (Schneider et al., 2019) generate continuous representations with powerful autoregressive models pre-trained to predict future time-step representations. Further works started discretizing the continuous representations with vq-VAE (van den Oord et al., 2017) to generate discrete representations for speech.

Works like vq-**wav2vec** (Baevski et al., 2019) and vq-APC (Chung et al., 2020) discretize the representations and perform the same prediction tasks as in **wav2vec** (Schneider et al., 2019) and APC (Chung & Glass, 2020) respectively but over discrete representations. In vq-**wav2vec**, the discrete representations are generated with either a K-Means Vector Quantizer (Baevski et al., 2019) or Gumbel-Softmax based Vector Quantizer (Baevski et al., 2019). The learned discrete representations are used to pre-train a BERT (Devlin et al., 2018) to generate stronger continuous representations much like BERT pre-training in Natural Language Processing. **wav2vec** 2.0 (Baevski et al.,

2020) uses a Gumble Softmax based Vector Quantizer (Baevski et al., 2019) to generate discrete representations. The training involves masking of spans of time steps and then predicting the correct discrete representations at each masked time step with transformer representation at that time step. In these methods, raw audio is discretized in a latent space to model all possible acoustic units than phonetic or sub-phonetic units. The tokens generated by the vector quantizers aren't constrained to be interpretable and are initialized in large numbers ($\sim 102.4K$ codes). After pre-training, a subset of these codes or tokens are chosen more often by the vector quantizers and are considered to represent acoustic units. CTC-based fine-tuning with transcription groups these discrete acoustic units to $K$ distinct phonemes or linguistic units as present in the transcriptions.

Works like HuBERT and **wav2vec**-Unsupervised learn phonemic units directly. HuBERT (Hsu et al., 2021) pre-trains a transformer network via BERT-like masked prediction task over noisy targets generated with a clustering model trained offline. The targets may be generated with an ensemble of $K$-means clusterers with $K = \{100, 500\}$ clusters on MFCC features or transformer representations. **wav2vec**-Unsupervised (Baevski et al., 2021) learns phonetic tokens adversarially from phonemized unlabelled text data. A discriminator identifies if the phoneme sequence generated by model is real or fake based on phonemized unlabelled text.

All aforementioned approaches use powerful auto-regressive models pre-trained on large amounts of unlabeled audio and fine-tuned on transcribed audio. Our learning approach can learn semantic tokens with small models while training pairwise on small amount of unlabelled audio data.

**Audio Representations for Retrieval.** Now Playing (Arcas et al., 2017) and (Chang et al., 2020) use a Neural Network Fingerprinter (NNFP) module outputting representations which are efficient for search in query-by-example tasks where the difference between query and the actual song is pretty minute in comparison to humming where only the melody is sung. Now Playing (Arcas et al., 2017) trains representations by optimizing the triplet loss (Schroff et al., 2015) and (Chang et al., 2020) trains representations by simulating the Maximum Inner Product Search (MIPS) on minibatches of representations. For Query by Humming task, (Mostafa et al., 2016) and (Mostafa & Fung, 2017) use deep learning models like DNNs and CNNs to generate representations which they map to MIDI-numbers or note tokens. Such works require note-transcribed data to train the models. For Spoken Term Detection task, approaches like (Zhang & Glass, 2009), (Rodriguez-Fuentes et al., 2014), (Lee et al., 2015), (Ram et al., 2018) convert audio to sequences of feature vectors and apply different variations of DTW based template matching to detect query in long utterances of speech which is time-consuming.

**Cross Domain Alignment.** Given a pair of semantically similar inputs for training, tasks such as visual question answering (text and image) and machine translation (text) involve learning an alignment. The alignment here is not ordered and the inputs may be from different modalities. Attention models have been used to find alignment between output entities and input regions (Yao et al., 2018). (Chen et al., 2020) use Gromov-Wasserstein distance between output and input entities to match them. However, there is no notion of tokens there, rather the salient entities in the input are represented as vectors in a graph.

**Graph Matching.** Graph Neural Networks (Gori et al., 2005) are used to generate embeddings for graphs. These embeddings are used to perform graph matching to find similarity of structured graphs (Li et al., 2019). However, they perform the matching jointly on the pair of inputs, rather than representing each input independently. This makes them unsuitable for the search problem at hand due to large run-time complexity. The distance metrics used for graph matching are based on edit distance (Li et al., 2019) and Wasserstein distance (Chen et al., 2020).

## 3 PROBLEM STATEMENT

We aim to map $\mathcal{X}$, a sequence of vectors, to $\tilde{\mathcal{T}}$, a sequence of discrete tokens from a finite alphabet $\mathbb{A}$, such that the similarity of sequences is preserved in the sense of edit distance. The length of sequence $\tilde{\mathcal{T}}$ may be less than or equal to that of the sequence $\mathcal{X}$. In other words, given a pair of similar sequences $(\mathcal{X}_i, \mathcal{X}_j)$ and sequence $\mathcal{X}_k$ which is not similar to either sequences in the pair, we want to map them to token sequences such that $ED(\tilde{\mathcal{T}}_i, \tilde{\mathcal{T}}_j)$ should be less than $\min\{ED(\tilde{\mathcal{T}}_i, \tilde{\mathcal{T}}_k), ED(\tilde{\mathcal{T}}_j, \tilde{\mathcal{T}}_k)\}$, where $ED(\cdot, \cdot)$ is the edit distance between two sequences.

## 4   Model Architecture

**wav2tok** is comprised of an encoder $f : \mathbb{X} \mapsto \mathbb{Z}$ which takes as input a temporal sequence of audio features $\mathcal{X} = [\mathbf{x}_t \in R^n; t \in [T]]$ of length $T$, where $\mathbf{x}_t$ is the feature vector at time step $t$, and outputs a sequence of L-2 normalised representations $\mathcal{Z} = [\mathbf{z}_t = f(\mathbf{x}_t) \in R^m; t \in [T]]$. The encoder is implemented as a 2-layer BiLSTM followed by an L-2 normalization layer. BiLSTMs summarise information in both directions and encode surrounding context.

A $K$-means vector quantizer network $g : \mathbb{Z} \mapsto \mathbb{T}$ then labels sequence $\mathcal{Z}$ at each time-step with tokens belonging to a finite $K$-element alphabet $\mathbb{A} = [K]$ and generates sequence of tokens $\mathcal{T} = [\tau_t = g(\mathbf{z}_t) \in \mathbb{A}; t \in [T]]$. Network $g$ vector quantizes input $\mathbf{z}_t$ with a codebook $E = \{\mathbf{e}_k \in \mathbb{Z}; k \in [K]\}$ comprised of $|\mathbb{A}| = K$ discrete representations which are cluster centroids in representation space $\mathbb{Z}$ and outputs token $\tau_t = \arg\max_k \mathbf{z}_t \cdot \mathbf{e}_k$. Note, here the dot product gives a cosine similarity score since both the vectors are L-2 normalized, as a result, $\mathbf{e}_k \in E$ closest to $\mathbf{z}_t$ is chosen as its discrete representation and index $k$ as it's token $\tau_t$. The $K$ discrete representations in network $g$ are trainable parameters.

A compressor $\mathcal{C}$ compresses sequence of tokens $\mathcal{T}$ to sequence $\tilde{\mathcal{T}}$ of length $\tilde{T} \leq T$ by deleting all consecutive repetitions of tokens. $\mathcal{C}$ also generates the corresponding compressed sequence $\tilde{\mathcal{Z}}$ of length $\tilde{T}$ by averaging representations $\mathbf{z}_t \in \mathcal{Z}$ over the consecutive tokens and L-2 normalising the averaged representation. Figure 1a presents an illustration demonstrating our model architecture.

## 5   Training

**wav2tok** is trained on pairs of sequences of audio features $(\mathcal{X}, \mathcal{X}')$ where the raw audio corresponding to $\mathcal{X}'$ is an augmented replica of that corresponding to $\mathcal{X}$. We apply either pitch shift or time stretch or both augmentations to raw audio to generate its augmented replica. $\mathcal{X}$ and $\mathcal{X}'$ may differ in sources as well, i.e. a different person may sing the recording corresponding to $\mathcal{X}'$.

The discrete representations in quantizer $g$ codebook $E$ are initialized as $K$ centroids obtained via offline $K$-means clustering over freshly initialized encoder-generated representations. Given $(\mathcal{X}, \mathcal{X}')$, encoder $f$ generates sequence of representations $\mathcal{Z}$ from input $\mathcal{X}$ and $\mathcal{Z}'$ from $\mathcal{X}'$. Quantizer $g$ generates a sequence of tokens $\mathcal{T}$ from input $\mathcal{Z}$ and $\mathcal{T}'$ from $\mathcal{Z}'$ via cosine similarity-based comparison with codebook vectors $e \in E$ initialized via offline clustering over freshly initialized representation space $Z$. Compressor $\mathcal{C}$ compresses sequence of tokens $\mathcal{T}$ to sequence $\tilde{\mathcal{T}}$ and $\mathcal{T}'$ to $\tilde{\mathcal{T}}'$.

We average all encoder-generated representations in pair $(\mathcal{Z}, \mathcal{Z}')$ which map to the same token, say $\tau$, to generate a prototype for $\tau$. We then perform a contrastive task where we compare the prototype with each of the $K$ discrete representations in codebook $E$ and increase its similarity with the discrete representation corresponding to $\tau$. We also increase the likelihood that **wav2tok** maps pair $(\mathcal{X}, \mathcal{X}')$ to the same token sequence via CTC framework to minimize $ED(\tilde{\mathcal{T}}, \tilde{\mathcal{T}}')$.

Our loss function is defined as,

$$\mathcal{L} = \mathcal{L}_m(\mathcal{X}, \mathcal{X}') + \alpha \mathcal{L}_{ctc}(\mathcal{X}, \tilde{\mathcal{T}}') + \beta \mathcal{L}_{ctc}(\mathcal{X}', \tilde{\mathcal{T}}) \tag{1}$$

where $\mathcal{L}_m$ is loss defined for contrastive task, $\mathcal{L}_{ctc}$ is the loss maximising aforementioned likelihood, and $\alpha, \beta$ are positive constants. We optimize this loss function in a manner similar to the Expectation Maximization algorithm. The clustering is used as the E-step to update the discrete representations in quantizer $g$ codebook, while gradient descent over $\mathcal{L}$ acts as the M-step.

**Contrastive Loss.** Let the set of unique tokens occurring in pair $(\tilde{\mathcal{T}}, \tilde{\mathcal{T}}')$ be $\mathcal{U} \subset [K], |\mathcal{U}| = K' \leq K$. We generate a list of token prototypes $\mathcal{P} = \{\mathbf{p}_\tau; \tau \in \mathcal{U}\}$ where $\mathbf{p}_\tau$ is L-2 normalised mean of representations in $\{\mathbf{z} \in \{\mathcal{Z}; \mathcal{Z}'\} : g(\mathbf{z}) = \tau\}$. Figure 1b presents an illustration demonstrating how we generate list of token prototypes $\mathcal{P}$.

Given $\mathbf{p}_\tau \in \mathcal{P}$, we perform a contrastive task to increase its similarity with discrete representation $\mathbf{e}_\tau \in E$. To compare $\mathbf{p}_\tau$ with the codebook, metrics such as cosine similarity and Euclidean distance could be used. However, we find that using the following parameterized score for this purpose gives better performance,

$$\mathbf{s}_{\tau,k} = \sigma(W \cdot (\mathbf{p}_\tau - sg(\mathbf{e}_k))) \in [0, 1] \tag{2}$$

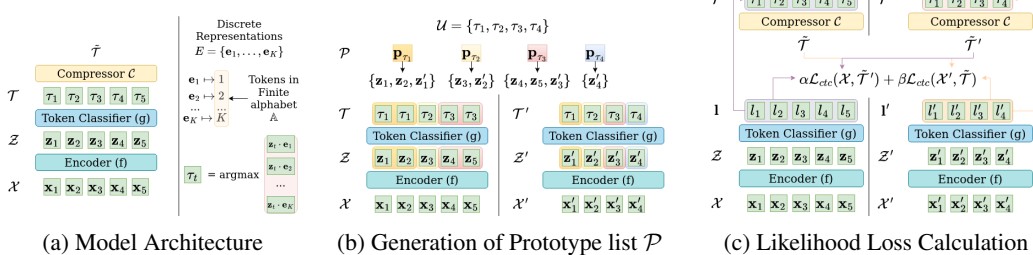

(a) Model Architecture  (b) Generation of Prototype list $\mathcal{P}$  (c) Likelihood Loss Calculation

Figure 1: $\mathcal{X}'$ is an augmented replica of $\mathcal{X}$. 1a illustrates our model architecture. 1b demonstrates the generation of $\mathcal{P}$ required for calculation of $\mathcal{L}_m$. 1c demonstrates our likelihood loss calculation.

where $sg(x) \equiv x, \frac{d}{dx}sg(x) \equiv 0$ is the stop-gradient operator, $\sigma(.)$ is sigmoid function generating a score in the range $[0, 1]$ and $W \in R^{1 \times d}$ is a parameter matrix. $\mathbf{s}_{\tau,k}$ acts as a parameterized similarity score between $\mathbf{p}_\tau$ and discrete representation $\mathbf{e}_k \in E$. We define our contrastive loss $\mathcal{L}_m$ as,

$$\mathcal{L}_m(\mathcal{X}, \mathcal{X}') = -\sum_{\tau \in \mathcal{U}} \log \frac{\exp(\mathbf{s}_{\tau,\tau})}{\sum_{k=1}^{K} \exp(\mathbf{s}_{\tau,k})} \tag{3}$$

**Likelihood Loss.** We maximize the likelihood that sequence $\mathcal{X}$ maps to token sequence $\tilde{\mathcal{T}}'$, which corresponds to $\mathcal{X}'$, via the CTC framework (see Figure 1c). It puts a constraint to generate the same token sequence for $\mathcal{X}$ and $\mathcal{X}'$. We calculate the probability of $\mathbf{x}_t$ mapping to token $\tau_t = k$ as $l_{t,k} = \frac{\exp(f(\mathbf{x}_t) \cdot sg(e_k))}{\sum_{i=1}^{K} \exp(f(\mathbf{x}_t) \cdot sg(e_i))}$. The likelihood $P(\tilde{\mathcal{T}}'|\mathcal{X})$ is then calculated as sum of probabilities of all $T$-length paths $\pi$ over tokens $\tau \in \mathbb{A}$ such that $\mathcal{C}(\pi) = \tilde{\mathcal{T}}'$. The loss is defined as,

$$\mathcal{L}_{ctc}(\mathcal{X}, \tilde{\mathcal{T}}') = -\log \sum_{\pi \in C^{-1}(\tilde{\mathcal{T}}')} P(\pi|\mathcal{X}) \tag{4}$$

where the path probabilities are calculated over token probability scores in sequence $\mathbf{l} = \{l_t \in R^K; t \in [T]\}$ via CTC forward-backward framework (Graves et al., 2006) without the use of blanks. We present the CTC forward and backward variables for our use case in Appendix B.

**Clustering.** We perform offline $K$-means clustering on a subset of encoder representations during initialization of our network and at regular intervals during training to set the discrete representations in codebook $E$ of network $g$. Initializing the clusters in this way prevents **wav2tok** from converging to a local optimum during the matching task, as is the case we found with random initialization of centroids. The intermittent clustering during training iteratively refines the discrete representations and prevents codebook collapse. We use the **sklearn** library to perform $K$-means clustering.

We train **wav2tok** using the ADAM (Kingma & Ba, 2017) optimizer and a linear learning schedule with a learning rate of $0.001$ and $8\%$ of the training steps as warm-up steps.

## 6 EXPERIMENTS

We test the performance of tokens and encoder-generated continuous representations of **wav2tok** in audio retrieval. We perform Query by Humming (QbH) and Spoken Term Detection experiments to evaluate the performance of **wav2tok** in comparison to the baselines.

### 6.1 MUSIC MELODY SEARCH: QUERY BY HUMMING

**Task.** Given a test query audio, we are to find the audio with the most similar melody in the search audio database.

**Experiment Details.** We use the MIR-QbSH dataset which is composed of $4431$ humming audio recordings of $30s$ duration corresponding to $48$ songs. Each song is sung by several individuals. All individuals sing the same part of the song. The recordings have variations in the environments

they were recorded in, tonal qualities, voices, pitch, and time stretch. We train our models on hums of 40 songs in MIR-QbSH dataset and evaluate search performance on hums of the remaining 8 songs. The training dataset has 1970 hums for training and 676 for validation. The test dataset has 225 hums as a search database and 659 query hums. We evaluate the performance of our models in identifying which song a given query corresponds to via comparison with all sequences in the search database. Each model converts all the audio in our test dataset to sequences of tokens or representations. Each query sequence is compared to all sequences in the search database via Edit Distance (ED) (if tokens) or DTW (if representations). The song id of the most similar sequence in the search database is then selected as query song id. We calculate Mean Reciprocal Ranking (MRR) score with ground-truth song id of the queries for evaluation. The Reciprocal Ranking (RR) score is given as $1/r$ if the $r^{th}$ most similar sequence in search database has same song id as query.

All the audio recordings are converted to Short Term Fourier Transform (STFT) matrices before being passed as inputs to our models. The STFT matrices are computed with 513 frequency bins, a window length of 1024 samples (summarising 128 ms of audio), and hop length of 512 samples.

## 6.2 SPOKEN TERM DETECTION

**Task.** Given a test query audio, we are to detect its occurrence in a long utterance.

**Experiment Details.** We use the TIMIT dataset which is composed of 6300 utterances of English speech with time-aligned word transcriptions. We choose 59 most-occurring words with more than 2 characters as keywords and all others as non-keywords. We use utterances of random sentences formed with 6 words sampled from a subset of 25 keywords for training and evaluation on STD experiments for the detection of the remaining 34 keywords. The test dataset is composed of 337 utterances corresponding to the 34 queries and 100 long utterances per query, with half containing a single occurrence of query amongst non-keywords and the other half containing only non-keywords. Given a query and a long utterance, we convert both to sequences of tokens using each audio tokenizer. We perform approximate string matching (Hall & Dowling, 1980) for detection of query in the utterance. The STFT matrix inputs to the models are computed with 185 frequency bins, a window length of 368 samples (summarising 23 ms of audio), and a hop length of 92 samples.

## 6.3 BASELINES

**Triplet.** We train encoder $f : \mathbb{X} \mapsto \mathbb{Z}$ to generate L-2 normalized continuous representations for retrieval. Encoder $f$ is trained via optimizing the triplet Loss (Schroff et al., 2015) as done in training an NNFP in Now Playing (Arcas et al., 2017). Given pair of similar sequences $(\mathcal{X}, \mathcal{X}')$, encoder $f$ generates sequences $\mathcal{Z}$ and $\mathcal{Z}'$. We form a mini-batch of size $N$ of triplets $\{\mathbf{z}, \mathbf{z}^+, \mathbf{z}^-\}$ where representation $\mathbf{z}$ is sampled from sequence $\mathcal{Z}$, $\mathbf{z}^+$ and $\mathbf{z}^-$ are positive and negative samples respectively for $\mathbf{z}$ sampled from sequence $\mathcal{Z}'$. The loss is defined as, $\mathcal{L}_{\textbf{Triplet}} = \sum_{i=1}^{N} max\{||\mathbf{z}_i - \mathbf{z}_i^+|| - ||\mathbf{z}_i - \mathbf{z}_i^-|| + m, 0\}$, where $m$ is a margin for similarity.

**MIPS.** We train encoder $f : \mathbb{X} \mapsto \mathbb{Z}$ to generate L-2 normalized continuous representations for retrieval. Encoder $f$ is trained via simulation of **MIPS** (Mussmann & Ermon, 2016) on mini-batches of representations as proposed by (Chang et al., 2020). Given pair of similar sequences $(\mathcal{X}, \mathcal{X}')$, encoder $f$ generates sequences $\mathcal{Z}$ and $\mathcal{Z}'$. We form a mini-batch of size $N$ of pairs of $\{\mathbf{z}, \mathbf{z}^+\}$ where encoder generated representation $\mathbf{z}$ is sampled from sequence $\mathcal{Z}$ and $\mathbf{z}^+$ is a positive for $\mathbf{z}$ sampled from $\mathcal{Z}'$. The loss is defined as, $\mathcal{L}_{\textbf{MIPS}} = -\sum_{i=1}^{N} \log \frac{\exp(\mathbf{z}_i, \mathbf{z}_i^+)}{\sum_{j \neq i} (\exp(\mathbf{z}_i \cdot \mathbf{z}_j^+) + \exp(\mathbf{z}_i \cdot \mathbf{z}_j))}$.

**wav2vec2.** We train our audio tokenizer via **wav2vec 2.0** (Baevski et al., 2020) learning framework. Quantizer $g$ in our audio tokenizer is chosen to be a Gumbel Softmax-based Vector Quantizer (See Appendix C for details) as used in (Baevski et al., 2020) but with a single codebook with $K$ members. Given sequence $\mathcal{X}$, encoder $f$ outputs sequence of L-2 normalised representations $\mathcal{Z}$ of length $T$. Quantizer $g$ outputs sequence of discrete representations $\mathcal{Q} = \{q_t = g(z_t \in \mathcal{Z}); t = 1, ..., T\}$. We mask spans of 10 time steps with random starting indices in sequence $\mathcal{Z}$ and then pass the new sequence to a transformer network $h : \mathbb{Z} \mapsto \mathbb{O}$ which generates a sequence of contextualized representations $\mathcal{O} = \{\mathbf{o}_t = h(z_t \in \mathcal{Z}); t = 1, ..., T\}$. For transformer output $\mathbf{o}_t$ over masked time step $t$, we identify the true discrete representation $\mathbf{q}_t$ from a set $\mathcal{D}_t$ composed of $\mathbf{q}_t$ and $D$ distractors which are discrete representations sampled from other time steps. The loss is defined as,

$\mathcal{L}_w(\mathbf{o}_t, \mathcal{D}_t) = -\log \frac{\exp(sim(\mathbf{o}_t, \mathbf{q}_t))}{\sum_{\tilde{q} \in \mathcal{D}_t} \exp(sim(\mathbf{o}_t, \tilde{q}))} + \mathcal{L}_d$, where $sim(a, b) = \frac{a^T b}{||a||||b||}$ is cosine similarity and $\mathcal{L}_d$ is codebook diversity loss.

**wav2vec2P.** We train **wav2vec2** audio tokenizer with our variation of **wav2vec** 2.0 (Baevski et al., 2020) learning framework which learns discrete representations from pairs of similar sequences. Given pair $(\mathcal{X}, \mathcal{X}')$, encoder $f$ outputs sequences $\mathcal{Z}$ of length $T$ and $\mathcal{Z}'$ of length $T'$ respectively. Assuming $T \leq T'$, we generate sequence $\mathcal{Z}^+$ of length $T$ whose $t$ time step element $\mathbf{z}_t^+$ is a positive for $\mathbf{z}_t \in \mathcal{Z}$ sampled from sequence $\mathcal{Z}'$. Gumbel Softmax-based Vector Quantizer $g$ quantizes each representation in sequence $\mathcal{Z}^+$ to generate sequence $\mathcal{Q}^+$. We mask sequence $\mathcal{Z}$ and $\mathcal{Z}^+$ at the same time steps. Transformer $h$ inputs masked sequences and generate sequences $\mathcal{O}$ and $\mathcal{O}^+$. For masked time step $t$, we use transformer output $\mathbf{o}_t$ to identify $\mathbf{q}_t^+ \in \mathcal{Q}^+$ from set $\mathcal{D}_t^+$ with distractors sampled from sequence $\mathcal{Q}^+$ and transformer output $\mathbf{o}_t^+$ to identify $\mathbf{q}_t \in \mathcal{Q}$ from set $\mathcal{D}_t$ with distractors sampled from sequence $\mathcal{Q}$. The loss is defined as, $\mathcal{L}_{wP} = \mathcal{L}_w(\mathbf{o}_t, \mathcal{D}_t^+) + \mathcal{L}_w(\mathbf{o}_t^+, \mathcal{D}_t)$.

**wav2vec2-O.** The original **wav2vec** 2.0 base model with 12 Transformer blocks and $95M$ parameters as proposed by (Baevski et al., 2020). It is pre-trained on 960 hours of LibriSpeech data and fine-tuned on TIMIT dataset. It uses $K = 32$ tokens for tokenization.

**wav2vec2-Multi.** A **wav2vec** 2.0 large model with 24 Transformer blocks and $317M$ parameters pre-trained on 53 languages as proposed by (Conneau et al., 2020). It is fine-tuned on Common Voice to detect all possible phonemes in training languages with $K = 392$ tokens.

**Triplet** and **MIPS** use a 2-layer BiLSTM as encoder with $3.6M$ parameters. We use the LAMB optimizer (You et al., 2020) and a Cosine Annealing Learning Schedule (Loshchilov & Hutter, 2017) with a learning rate restart of $0.0001$ to train them. **wav2vec2** and **wav2vec2**P use a 2-layer BiLSTM encoder with $3.6M$ parameters to generate latent representations and 3 Transformer blocks with $8.5M$ parameters. Both are trained using the ADAM (Kingma & Ba, 2017) optimizer and a linear learning schedule with a learning rate of $0.001$ and $8\%$ of the training steps as warm-up steps. Proposed **wav2tok** uses only a 2-layer BiLSTM as encoder with $3.6M$ parameters.

# 7 RESULTS

## 7.1 MUSIC MELODY SEARCH: QUERY BY HUMMING

We present search performances for 3 settings of query namely- Query with no augmentation or Vanilla Query (V), Time Stretched Query (TS), and Pitch Shifted Query (PS). Time stretch and pitch shift are the most common augmentations that may be faced in queries by humming data. No augmentations were applied to audio in search database. Evaluations are performed on sequences corresponding to songs not seen during training. The results present the generalizability of the tokens or representations generated by the models. We set the number of tokens as $K = 25$ for **wav2tok**, **wav2vec2**, and **wav2vec2**P (See Appendix A.2 for experiments to support our choices).

**Quality of Tokenization.** Table 1 presents the performance of the sequence of tokens $\tilde{\mathcal{T}}$ generated by the audio tokenizers on ED-based similarity search. Tokens generated by **wav2tok** present good generalization capabilities in terms of MRR and outperform all the baselines. It generates time and pitch invariant tokens as we see no drop in performance when either augmentation is applied to query. **wav2vec2**-O is trained on English speech only. The tokens generated by it do not contain much melodic information but are robust to augmentations. The multilingual training of **wav2vec2**-Multi infuses both melodic and phonetic information to its 392 tokens, thereby giving good performance. **wav2tok** outperforms both **wav2vec2**-O and **wav2vec2**-Multi given its pairwise training which allows it to infuse more melodic information to the tokens while also being trained on a small amount of unlabelled data. The Gumbel Softmax-based quantizer in **wav2vec2** and **wav2vec2**P isn't ideal for infusing melodic information to tokens but it does infuse phonetic information as will be seen in Section 7.2. We compare the tokens with representations learned by **MIPS** and **Triplet** evaluated on DTW-based similarity search. The continuous representations present sub-par generalizations to unseen songs. We compare **wav2tok** with SOTA melody extraction algorithm proposed in (Salamon & Gómez, 2012) which converts hums to **MIDI** sequences. **wav2tok** generates token sequences much smaller than the respective **MIDI** sequences and outperforms the **MIDI** tokens in search performance, search time, and robustness. In addition, **wav2tok** outperforms the algorithm in inference time. We further compare **wav2tok** with SOTA QbH system proposed in (Mostafa &

Fung, 2017). In our implementation, we map audio to **MIDI** sequences using the aforementioned SOTA melody extraction algorithm instead of a CNN. Given **MIDI** sequence $53, 53, 58, 50$ with durations $0s, 0.5s, 1s, 2s$, a **Relative Note** sequence is generated as $(0,0), (0, 0.5), (5, 1), (-8, 2)$ over which DTW is performed for retrieval. **wav2tok** tokens outperform the SOTA QbH system in both performance and robustness; the performance of the latter drops drastically with time stretch.

We present the performances of the uncompressed sequences $\mathcal{T}$ and $\mathcal{Z}$ and compressed sequence $\tilde{\mathcal{Z}}$ generated by the audio tokenizers in Appendix A.1. We observe a drop in performance for all audio tokenizers when we apply sequence compression to sequences $\mathcal{T}$ and $\mathcal{Z}$. **wav2tok** outperforms all the baselines and generates superior-quality of continuous representations and discrete tokens.

**Search Time.** Table 1 presents the search time taken for similarity search over the tokens or representations generated by the models. The search time taken per query is 2 order of magnitude lesser for ED-based Search over compressed sequence of tokens $\tilde{\mathcal{T}}$ than standard DTW-based Search over continuous representations $\mathcal{Z}$. The pre-trained models being fine-tuned on transcribed audio give the best tokens in terms of compression and search time. **wav2tok** gives comparable tokens but outperforms the pre-trained models in inference time.

Table 1: Quality of Tokenization

Table 2: Ablation Studies and Some Variations

| Model | V (MRR) | TS (MRR) | PS (MRR) | Search Time (s) | Infer (s) |
|---|---|---|---|---|---|
| **MIDI** ED | 0.75 | 0.64 | 0.72 | 3.84 | 0.62 |
| **Relative Note** DTW | 0.84 | 0.74 | 0.8 | 0.02 | 0.62 |
| **Triplet** DTW | 0.5 | 0.48 | 0.5 | 3.5 | 0.1 |
| **MIPS** DTW | 0.6 | 0.55 | 0.58 | | |
| **wav2vec2** ED | 0.66 | 0.63 | 0.64 | 0.06 | 0.17 |
| **wav2vec2**P ED | 0.69 | 0.65 | 0.67 | | |
| **wav2vec2**-O ED | 0.72 | 0.72 | 0.71 | 0.01 | 0.43 |
| **wav2vec2**-Multi ED | 0.82 | 0.82 | 0.82 | | 1.2 |
| **wav2tok** ED | **0.84** | **0.84** | **0.84** | 0.04 | 0.14 |

| Model | V (MRR) | TS (MRR) | PS (MRR) |
|---|---|---|---|
| **log-mel** DTW | 0.72 | 0.7 | 0.67 |
| vq-**log-mel** ED | 0.71 | 0.6 | 0.62 |
| **wav2tok**+NoSim ED | 0.73 | 0.73 | 0.72 |
| **wav2tok**+Cos ED | 0.79 | 0.76 | 0.77 |
| **wav2tok**+CTC ED | 0.64 | 0.62 | 0.63 |
| **wav2tok**+NewInit ED | 0.77 | 0.76 | 0.78 |
| **wav2tok**+MIR1K ED | 0.72 | 0.64 | 0.67 |
| **wav2tok** ED | **0.84** | **0.84** | **0.84** |

**Ablation Studies.** Query by humming involves similarity based on melody information, which is carried by the semantic pairing of the audio in training data. We constrain this pairing to include sequences not semantically similar and call this model **wav2tok**+NoSim. We optimize the contrastive loss $\mathcal{L}_m$ to train the model. The results are shown in Table 2 (full table in Appendix A.3). There is a significant drop in token robustness and performance but the representations suffer a small drop (see Appendix A.3). Hence, although the representation space may be well clustered, **wav2tok** is able to add more semantics to the tokens as it is being trained with pairs of similar sequences in comparison to **wav2tok**+NoSim. We train **wav2tok** with cosine similarity scores instead of a parameterized score (**wav2tok**+Cos). The drop in performance validates the enhancement brought about by using a parameterized score. We also train **wav2tok** with $\mathcal{L}_{ctc}$ only (**wav2tok**+CTC). The CTC loss considers all possible paths which compress to the target label sequence. As a result, the learnt tokens aren't much semantic. The use of both losses gives the best tokens.

**Some Variations.** In **wav2tok**+NewInit, we associate the discrete representations with $K$ centroids in the input space $\mathbb{X}$. Such association does not initialize our tokenizer with optimal centroids which cluster the space $\mathbb{Z}$ perfectly. This results in a significant drop in performance and robustness as shown in Table 2. We train **wav2tok** on MIR-1K dataset (**wav2tok**+MIR1K) which is composed of polyphonic music recordings of 1000 distinct songs. The tokens generalize well to monophonic hums in MIR-QbSH dataset giving a comparable performance to **MIDI** tokens. This validates that **wav2tok** tokens do learn melodic information and are robust to variations incurred in hums. We further compare **wav2tok** with **log-mel** features and token sequences (with no compression) obtained via quantization of **log-mel** features. **wav2tok** tokens outperform both.

## 7.2 SPOKEN TERM DETECTION

**Quality of Tokenization.** Table 3 presents the quality of tokenization of the query keywords by the models evaluated in the Spoken Term Detection experiments. We present the performances of **wav2vec2** , **wav2vec2**P, **wav2vec2**-O, **wav2vec2**-Multi and proposed **wav2tok**. We conduct search experiments on a test dataset composed of a search database of 337 utterances of the 34 keywords used as queries in the STD experiments and 1289 query utterances. We identify the keyword to

which each query corresponds to via comparison to all the 337 utterances in the search database via ED-based similarity score. The word id of the most similar utterance is selected as the word to which the query corresponds to. We set $K = 40$ equivalent to the number of phonemes in English.

**wav2tok** gives the best performance in terms of MRR scores. It outperforms huge models like **wav2vec2**-O and **wav2vec2**-Multi which are fine-tuned for the task of phonetic tokenization of speech audio while using a small number of parameters. **wav2vec2** and **wav2vec2**P also outperform **wav2vec2**-Multi and **wav2vec2**-O while using smaller number of parameters. **wav2vec2**-O and **wav2vec2**-Multi use a blank token to handle consecutive occurrences of the same tokens and to label background noise. The utterances of each keyword in the test dataset are very small in time duration. This causes **wav2vec2**-O to confuse word utterances as background noise. It generates a sequence of blank tokens and performs poorly in search. **wav2vec2**-Multi using a larger number of phonetic tokens does not suffer this issue. **wav2tok** , **wav2vec2**, and **wav2vec2**P have no such blank token. This brings a drop in search performance with sequence compression. We further present the performance of **wav2tok** trained on a much larger LibriSpeech 100 hours dataset (**wav2tok**+Libri). It is able to outperform **wav2vec2**-O and give comparable performance to **wav2vec2**-Multi.

Table 3: Quality of Tokenization for speech

| Model | Normal (MRR) | Compressed (MRR) |
|---|---|---|
| **log-mel** DTW | 0.7 | - |
| **wav2vec2** ED | 0.68 | 0.63 |
| **wav2vec2**P ED | 0.7 | 0.65 |
| **wav2vec2**-O ED | 0.4 | 0.4 |
| **wav2vec2**-Multi ED | 0.67 | 0.67 |
| **wav2tok** ED | **0.74** | 0.66 |
| **wav2tok**+Libri ED | 0.64 | 0.6 |

Table 4: Spoken Term Detection

| Model | ED (F1) | Search Time (s) | DTW (F1) | Search Time (s) |
|---|---|---|---|---|
| **log-mel** DTW | - | - | 0.41 | 0.003 |
| **wav2vec2** | 0.64 | 0.066 | 0.46 | 0.1 |
| **wav2vec2**P | 0.64 | | 0.47 | |
| **wav2vec2**-O | 0.61 | 0.29 | 0.43 | 0.23 |
| **wav2vec2**-Multi | 0.63 | 0.72 | 0.48 | 0.66 |
| **wav2tok** | **0.65** | **0.064** | **0.52** | **0.09** |
| **wav2tok**+Libri | 0.63 | | 0.44 | 0.1 |

**Spoken Term Detection.** We convert the query word utterance and the long utterance in to sequences of tokens by all our models and detect the occurrence of the query via approximate string matching. We use **fuzzysearch** library to perform approximate string matching. It automatically chooses the fastest algorithm for matching. Table 4 presents the performance of **wav2vec2** , **wav2vec2**P , **wav2vec2**-O, **wav2vec2**-Multi, and proposed **wav2tok** in STD. All the models give a comparable performance in terms of F1- score with **wav2tok** performing slightly better. We also implement the STD system proposed in (Anguera & Ferrarons, 2013) which performs highly competitive STD via subsequence DTW (S-DTW) over gaussian posterior features. In our implementation, we extract the posterior features with SOTA ASR models like **wav2vec2**-O and **wav2vec2**-Multi. The results are presented in the DTW column in Table 4. Note, the results for other models in same column are for STD via S-DTW over representations. We observe STD over tokens to give better F1-score.

## 8 CONCLUSION AND FUTURE WORK

In this paper, we present an audio sequence tokenizer **wav2tok** that generates semantically meaningful ordered representations (or tokens) that can be used for efficient retrieval by query sequences. The model learns only from pairs of semantically similar sequences and outperforms state-of-the-art approaches for spoken term detection and query by humming. One may apply more efficient search algorithms such as locality-sensitive hashing and longest common subsequence search on the generated tokens to further speed up the search. The proposed framework can also be extended to image and video retrieval, as they also have spatial ordering. We would like to investigate the domain-specific, i.e., linguistic or musicological, aspects of the extracted tokens. For instance, during retrieval, the matching algorithm assumes all the tokens to be equidistant from each other. One may study or use the metric space of these tokens.

## 9 REPRODUCIBILITY

The codes are available in `https://github.com/madhavlab/wav2tok`. The experiments are performed using standard datasets.

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

## A    FURTHER STUDIES

### A.1    SEQUENCE COMPRESSION

We present the quality of sequence of tokens $\mathcal{T}$ and sequence of representations $\mathcal{Z}$ and their corresponding compressed versions sequences $\widetilde{\mathcal{T}}$ and $\widetilde{\mathcal{Z}}$ generated by the audio tokenizers in Table 5.

**wav2tok** outperformed the baselines and generated the best quality of sequences $\mathcal{T}$, $\mathcal{Z}$, $\widetilde{\mathcal{T}}$ and $\widetilde{\mathcal{Z}}$. Sequence compression brings an order of magnitude drop in search time for all the audio tokenizers with a trade-off in search performance. Compression from $\mathcal{T}$ to $\widetilde{\mathcal{T}}$ increases the robustness of the token sequences generated by **wav2tok** to various augmentations. **wav2vec2**P learnt better tokens and representations than **wav2vec2** because of it's pairwise training on similar audio.

Table 5: Compression of Sequences: MRR scores for query by humming, K = 25

| Model | V | TS | PS | Search Time |
|---|---|---|---|---|
| **Without Compression** | | | | |
| **wav2vec2** DTW | 0.85 | 0.84 | 0.84 | |
| **wav2vec2**P DTW | 0.87 | 0.85 | 0.87 | 3.5s |
| **wav2tok** DTW | **0.92** | **0.89** | **0.93** | |
| **wav2vec2** ED | 0.72 | 0.69 | 0.73 | 0.68s |
| **wav2vec2**P ED | 0.75 | 0.71 | 0.73 | |
| **wav2tok** ED | **0.9** | **0.84** | **0.9** | **0.32s** |
| **With Compression** | | | | |
| **wav2vec2** DTW | 0.76 | 0.72 | 0.74 | 0.8s |
| **wav2vec2**P DTW | 0.81 | 0.77 | 0.79 | |
| **wav2tok** DTW | **0.88** | **0.88** | **0.87** | **0.6s** |
| **wav2vec2** ED | 0.66 | 0.63 | 0.64 | 0.06s |
| **wav2vec2**P ED | 0.69 | 0.65 | 0.67 | |
| **wav2tok** ED | **0.84** | **0.84** | **0.84** | **0.04s** |

### A.2    VARIATION IN NUMBER OF TOKENS $K$

The effect of varying the size of alphabet $\mathbb{A}$ is shown in Table 6. We train **wav2vec2**, **wav2vec2**P, and proposed **wav2tok** with alphabets of size $K \in \{15, 25, 40\}$. Out of the three settings for $K$, $K = 25$ gives the best performance for all models. **wav2tok** gives best performance for all settings of $K$.

### A.3    ABLATION STUDIES AND SOME VARIATIONS

We present the full version of Table 2 in table 7. Note **wav2tok**+NoSim repsentations are well clustered. **wav2tok**+Trans representations are also comparable with **wav2tok** but the tokens are of lesser quality. This is due to model overfitting.

### A.4    QUALITY OF REPRESENTATIONS

We present the performance of the continuous representations generated by **wav2tok** and the baselines in Table 8. **wav2tok** generates the best representations for music outperforming representations generated by the large **wav2vec** 2.0 models. **wav2tok** trained on MIR1K generates representations outperforming domain-specific QbH baselines. Note, **wav2vec2**-O outperforms **wav2vec2**-Multi as the hums in the dataset were all in english. **wav2vec2**-O is pre-trained and fine-tuned on English speech only while **wav2vec2**-Multi is pre-trained multilingually. Hence , **wav2vec2**-O gave better results.

Table 6: Effect of varying $K$: MRR scores for query by humming

| Models | Without Compression | | | With Compression | | |
|---|---|---|---|---|---|---|
| | V | TS | PS | V | TS | PS |
| $K = 15$ | | | | | | |
| **wav2vec2** DTW | 0.85 | 0.83 | 0.84 | 0.7 | 0.66 | 0.67 |
| **wav2vec2**P DTW | 0.87 | 0.85 | 0.85 | 0.82 | 0.77 | 0.8 |
| **wav2tok** DTW | **0.88** | **0.87** | **0.88** | **0.84** | **0.8** | **0.83** |
| **wav2vec2** ED | 0.79 | 0.77 | 0.78 | 0.58 | 0.56 | 0.57 |
| **wav2vec2**P ED | 0.8 | 0.77 | 0.79 | 0.71 | 0.68 | 0.7 |
| **wav2tok** ED | **0.82** | **0.8** | **0.81** | **0.77** | **0.75** | **0.76** |
| $K = 25$ | | | | | | |
| **wav2vec2** DTW | 0.85 | 0.84 | 0.84 | 0.76 | 0.72 | 0.74 |
| **wav2vec2**P DTW | 0.87 | 0.85 | 0.87 | 0.81 | 0.77 | 0.79 |
| **wav2tok** DTW | **0.92** | **0.89** | **0.93** | **0.88** | **0.88** | **0.87** |
| **wav2vec2** ED | 0.72 | 0.69 | 0.73 | 0.66 | 0.63 | 0.64 |
| **wav2vec2**P ED | 0.75 | 0.71 | 0.73 | 0.69 | 0.65 | 0.67 |
| **wav2tok** ED | **0.9** | **0.84** | **0.9** | **0.84** | **0.84** | **0.84** |
| $K = 40$ | | | | | | |
| **wav2vec2** DTW | 0.84 | 0.82 | 0.83 | 0.72 | 0.68 | 0.7 |
| **wav2vec2**P DTW | 0.86 | 0.85 | 0.85 | 0.81 | 0.77 | 0.79 |
| **wav2tok** DTW | **0.9** | **0.88** | **0.89** | **0.86** | **0.83** | **0.83** |
| **wav2vec2** ED | 0.71 | 0.66 | 0.69 | 0.6 | 0.58 | 0.58 |
| **wav2vec2**P ED | 0.73 | 0.7 | 0.73 | 0.68 | 0.65 | 0.67 |
| **wav2tok** ED | **0.83** | **0.8** | **0.82** | **0.77** | **0.75** | **0.76** |

Table 7: Ablation Studies and Some Variations: MRR scores for query by humming

| Models | Without Compression | | | With Compression | | |
|---|---|---|---|---|---|---|
| | V | TS | PS | V | TS | PS |
| **log-mel** DTW | 0.72 | 0.69 | 0.67 | 0.54 | 0.47 | 0.43 |
| **wav2tok**+NoSim DTW | 0.88 | 0.87 | 0.87 | 0.8 | 0.84 | 0.83 |
| **wav2tok**+Cos DTW | 0.88 | 0.87 | 0.87 | 0.83 | 0.81 | 0.81 |
| **wav2tok**+NewInit DTW | 0.9 | 0.84 | 0.91 | 0.84 | 0.85 | 0.83 |
| **wav2tok**+Trans DTW | 0.84 | 0.77 | 0.85 | 0.8 | 0.77 | 0.76 |
| **wav2tok**+MIR1K DTW | 0.88 | 0.84 | 0.85 | 0.82 | 0.74 | 0.78 |
| **wav2tok** DTW | **0.92** | **0.89** | **0.93** | **0.88** | **0.88** | **0.87** |
| vq-**log-mel** ED | 0.71 | 0.6 | 0.62 | 0.52 | 0.48 | 0.47 |
| **wav2tok**+NoSim ED | 0.85 | 0.74 | 0.84 | 0.73 | 0.73 | 0.72 |
| **wav2tok**+Cos ED | 0.86 | 0.84 | 0.85 | 0.79 | 0.76 | 0.77 |
| **wav2tok**+NewInit ED | 0.83 | 0.72 | 0.85 | 0.77 | 0.76 | 0.78 |
| **wav2tok**+Trans ED | 0.84 | 0.77 | 0.85 | 0.7 | 0.66 | 0.67 |
| **wav2tok**+MIR1K ED | 0.76 | 0.66 | 0.71 | 0.72 | 0.64 | 0.67 |
| **wav2tok** ED | **0.9** | **0.84** | **0.9** | **0.84** | **0.84** | **0.84** |

## A.5 TRAINING ON LARGER SPEECH DATASET

We train **wav2tok** on 100-hours subset of LibriSpeech (Panayotov et al., 2015) dataset. We evaluate the quality of tokenization of word utterances done by **wav2tok** on TIMIT (Garofolo et al., 1993) dataset. We use a 2-layer BiLSTM network with 3.6 million parameters as encoder network which takes MFCC feature sequences as input. We perform tokenization with $K = 40$ tokens.

**wav2tok** outperforms **wav2vec2**-O by a large margin and gives comparable performance to **wav2vec2**-Multi in terms of MRR score. **wav2tok** uses a minute number of parameters in comparison to 95 million parameters in **wav2vec2**-O and 317 million parameters in **wav2vec2**-Multi. Note, **wav2vec2**-O and **wav2vec2**-Multi were pre-trained on large amount of unlabelled speech data and

Table 8: Quality of Representations: MRR scores for query by humming

| Model | V | TS | PS |
|---|---|---|---|
| (Salamon & Gómez, 2012) **MIDI** ED | 0.75 | 0.64 | 0.72 |
| (Mostafa & Fung, 2017) Note DTW | 0.84 | 0.74 | 0.8 |
| **Triplet** DTW | 0.5 | 0.48 | 0.5 |
| **MIPS** DTW | 0.6 | 0.55 | 0.58 |
| **wav2vec2**-O DTW | 0.91 | 0.83 | 0.86 |
| **wav2vec2**-Multi DTW | 0.88 | 0.83 | 0.85 |
| **wav2tok** DTW | **0.92** | **0.9** | **0.93** |
| **wav2tok**+MIR1K DTW | 0.88 | 0.84 | 0.85 |

fine-tuned with transcription to perform tokenization of audio. Moreover, **wav2vec2**-O was fine-tuned to perform tokenization on TIMIT (Garofolo et al., 1993) dataset. Proposed **wav2tok** was trained on 100 hours of LibriSpeech dataset only. The tokens learnt by **wav2tok** on LibriSpeech (Panayotov et al., 2015) dataset generalised well to TIMIT (Garofolo et al., 1993).

Table 9: Quality of Tokenization for speech (MRR Scores)

| Model | Normal ($\mathcal{T}$) | Compressed ($\widetilde{\mathcal{T}}$) |
|---|---|---|
| **wav2vec2**-O ED | 0.4 | 0.4 |
| **wav2vec2**-Multi ED | 0.67 | 0.67 |
| **wav2tok**+Libri ED | 0.64 | 0.6 |

# B CTC WITHOUT BLANKS

We present the forward and backward variables used in calculating the gradients of the CTC loss $\mathcal{L}_{ctc}(\mathcal{X}, \tilde{\mathcal{T}}')$ with no blank tokens.

The forward variable is defined as ,

$$\alpha_t(s) = \sum_{\pi;\mathcal{C}(\pi_{1:t})=\tilde{\mathcal{T}}'_{1:s}} \prod_{t'=1}^{t} l_{t',\pi_{t'}} \tag{5}$$

where $\pi$ corresponds to all $T$-length paths over tokens such that $\mathcal{C}(\pi) = \tilde{\mathcal{T}}'$. Here, $\mathcal{C}$ is a compressor which compresses $\pi$ a $T$-length sequence of tokens via de-duplication.

We initialise as follows,

$$\begin{aligned} \alpha_1(1) &= l_{1,\tilde{\mathcal{T}}_1'} \\ \alpha_1(s) &= 0, \forall s > 1 \end{aligned} \tag{6}$$

and recursively calculate $\alpha_t(s)$ as,

$$\alpha_t(s) = (\alpha_{t-1}(s) + \alpha_{t-1}(s-1))l_{t,\tilde{\mathcal{T}}_s'} \tag{7}$$

We set $\alpha_t(s) = 0, \forall s < 1$.

The backward variable is defined as,

$$\beta_t(s) = \sum_{\pi;\mathcal{C}(\pi_{t:T})=\tilde{\mathcal{T}}'_{s:|\tilde{\mathcal{T}}'|}} \prod_{t'=t}^{T} l_{t',\pi_{t'}} \tag{8}$$

We initialise as follows,

$$\beta_T(|\tilde{\mathcal{T}}'|) = l_{T,\tilde{\mathcal{T}}'_{|\tilde{\mathcal{T}}'|}}$$
$$\beta_T(s) = 0, \forall s < |\tilde{\mathcal{T}}'| \tag{9}$$

and recursively calculate $\beta_t(s)$ as,

$$\beta_t(s) = (\beta_{t+1}(s) + \beta_{t+1}(s+1))l_{t,\tilde{\mathcal{T}}'_s} \tag{10}$$

We set $\beta_t(s) = 0, \forall s > |\tilde{\mathcal{T}}'|$.

## C  GUMBEL SOFTMAX BASED VECTOR QUANTIZER

The Gumbel Softmax based Vector Quantizer (Baevski et al., 2019) quantizes input latent representation $z_t \in R^m$ with $C$ codebooks containing $K$ quantizers $e \in R^{K \times \frac{m}{C}}$ each. For our experiments, we set $C = 1$ and $K \in \{15, 25, 40\}$. Given $\mathbf{z}_t$, one of the $K$ quantizers from each of the $C$ codebooks are chosen resulting in vectors $e_1, ..., e_C$. The codebook vectors are then concatenated and linearly transformed from $R^m$ to $R^d$ to output a discrete representation $q_t \in R^d$.

$\mathbf{z}_t$ is mapped to $\mathbf{l} \in R^{C \times K}$ logits to give probability scores for the choice of codeword. The probability $p_{c,k}$ of choosing $k^{th}$ quantizer in $c^{th}$ codebook is given as,

$$p_{c,k} = \frac{\exp\left(l_{c,k} + n_k\right)/\tau}{\sum_{i=1}^K \exp\left(l_{c,i} + n_i\right)/\tau} \tag{11}$$

where $\tau$ is a non-negative temperature, $n = -log(-log(u))$ and $u$ are samples from the uniform distribution **Unif**$(0, 1)$.

During forward pass, the codeword is chosen as $\kappa = \arg\max_j p_{c,j}$. During backward pass, the loss is calculated over the gumble softmax distribution $p$. We use the straight-through gradient estimator (Yin et al., 2019) to estimate the gradient.

**Codebook Diversity Loss** $\mathcal{L}_d$. This loss promotes equal use of all the entries in each of the $C$ codebooks. Minimization of this loss maximizes the entropy of the averaged softmax distribution $\tilde{p}$ over the $K$ entries for each codebook $\tilde{p}_c$ across a batch of utterances.

$$\mathcal{L}_d = \frac{1}{CK} \sum_{c=1}^C \sum_{k=1}^K \tilde{p}_{c,k} \log \tilde{p}_{c,k} \tag{12}$$

