# OpenReview forum: "wav2tok: Deep Sequence Tokenizer for Audio Retrieval"
_ICLR.cc/2023/Conference — ICLR 2023 poster_

### Official Review · Reviewer_LKSS · 2022-10-24

**Confidence:** 4
**Correctness:** 2
**Technical Novelty And Significance:** 3
**Empirical Novelty And Significance:** 2
**Recommendation:** 6

**Clarity, Quality, Novelty And Reproducibility:**

Clarity:

* Are the MIP and triplet baselines the systems described by Arcas et al (2017) and Chang et al (2020) or are there differences? This should be clarified.
* While page 3 states that neural network fingerprinter approaches are unsuited to comparing fully produced tracks with user query hums in query by humming, this does not appear to be the task that is evaluated in the experiments. The experiments focus on the more straightforward task of matching two hums together, for which NNFP may be well suited.
* Please use \citep and \citet to cite papers either in parentheses (Arcas et al., 2017) or textually as Arcas et al. (2017) say...
* Data augmentations are shown in figure 2, but it is never stated which augmentations are used in the main body of the paper.
* Approximate string matching is mentioned on page 6, but no specific algorithm or citation is provided.
* The description of the wav2vec baselines should be clarified to highlight what exactly is different between them. Presumably it is mainly the data used to train them, but this is not clearly stated.
* The dataset for the speech queries is not clearly explained. How were the target words selected? How were the sentences selected from TIMIT? How were they put together? Were there artifacts from this combination process? Does each sentence in the database just have a single target word or could any word in it be targeted? How does this task compare to the others mentioned in the introduction?
* Wav2Tok+Cos is mentioned on page 8, but is not present in table 2
* "Locally sensitive hashing" -> "Locality sensitive hashing"

Novelty:
* The approach appears to be novel

Reproducibility:
* The authors state that they will release the code after the review process, although there is no way to ensure this.

**Strength And Weaknesses:**

Strengths:
* The proposed approach seems to work well on the proposed tasks and datasets
* The idea of using the CTC from one sequence with the labels of a different sequence (and vice versa) is clever and interesting.

Weaknesses:
* Scalability. Fundamentally, the proposed approach takes time linear in the size of the database being searched to perform a single query. In general, it is difficult to achieve sub-linear matching speed in the size of a database for sequence matching.
* Need for sequence matching. Given questions about the speed of sequence matching, I wonder about the need for it. A clear comparison with a system that does not require sequence matching demonstrating that the sequence-based approach is superior would resolve this. For example, the proposed system could be compared with a similar system that does matching based on a fixed length representation.
* Need for discrete tokens. It is also not clear why discrete tokens are necessary. wav2vec 2.0, which uses discrete tokens has evolved into data2vec, which uses continuous hidden representations as targets of prediction. The reason that discrete tokens are preferable to a continuous representation in this case are not clear.
* Summarization of tokens. The idea of the replacement of a sequence of the same token with a single instance of it was introduced, but it is not clear why it is necessary. It is not shown in figure 1 and page 8 states "We observe a drop in performance for all audio tokenizers when we apply sequence compression to sequences T and Z." So is it used? Is it necessary?
* Baselines could be better. In addition to the fixed-length encoders above, it would be good to compare with DTW on mel spectrograms, and perhaps one with edit distance-based matching on k-means quantized mel spectrograms.
* No citations are provided in the introduction. While many citations are provided later in the paper, they would be very helpful to the reader in the introduction to support various claims.

**Summary Of The Paper:**

This paper describes a system that learns to encode audio files into sequences of discrete tokens for the purposes of performing sequence matching to a database of such sequences. The approach is evaluated in terms of matching query by humming queries to one another on a dataset of 48 songs each hummed about 10 times as well as a word search experiment with 34 keywords spoken in 337 utterances queried by 1289 queries. In comparison with a number of neural baselines, the proposed approach provided more accurate matches.

**Summary Of The Review:**

Overall, this approach is not especially well motivated, but does appear to work better than several baselines (although these are perhaps not the best baselines) on two experiments involving rather small datasets. Several important details of the approach are not entirely clear.

---

> ### Author Response · Authors · 2022-11-17
> **The results are for compressed token sequences. We didn't state NNFP is not well suited. We have added more competitive baselines. Added details to Spoken term dataset creation.**
>
> We thank the reviewer for the insightful comments. We address each of the questions and concerns as follows,
>
>      1. Scalability. .... for sequence matching.
>
>
> Response:
>
> We have presented the quality of representations generated by Wav2Tok in Appendix A.4 in the revised version of our manuscript.
>
> The representations generated are state-of-the-art.
> We can use LSH over the representations to attain speed up in search.
>
>
>
>     2. Need for sequence matching .... representation.
>
>
> Response:
>
>  We don't know the word boundaries for STD and QbH tasks. Hence, we can't use fixed length representations when our database has long audio recordings from which the query matches a small segment only.
>
>
>
>     3. Need for discrete tokens....not clear.
>
>
> Response:
>
> Discrete tokens can be compressed (de-duplication) and lead to faster and memory-efficient retrieval.
>
>
>
>        4. Summarization of tokens.... Is it necessary?
>
>
> Response:
>
> Tables 1 and 2 presents the performances of token sequences T when sequence compression is applied to it.
>
> In appendix A.1 Table 5 we have provided the results we attained with sequences T and Z and their respective compressed sequences.
> Note the drop in search time which we attained with compression.
>
> Sequence compression leads to faster search with a trade-off in performance.
>
>
>
>     5. Baselines could be better.... mel spectrograms.
>
>
> Response:
>
> Thanks for this comment.
> Based on this comment , we have added
>
>
> - DTW on mel Spectrograms
> - ED based matching on k-means quantized mel spectrograms
>
>
> as  baselines. The results are added to table 2 in the revised version of our manuscript. The results are discussed in last $2$ lines of section $7.1$, Some Variations paragraph.
>
> Based on the review by 7rFd, we have added the following baselines as well,
>
> Query by Humming:
>
>
>  - a SOTA melody extraction algorithm by Salamon et al, 2012. which maps audio to midi sequences.
> - a SOTA query-by-humming (QbH) system proposed by Mostafa and Fung, 2017
>
> The results are added to Table 1 in the revised version of our manuscript.
> The results are discussed in the last $7$ lines of section $7.1$, Quality of Tokenization paragraph.
>
> Spoken Term Detection
>
>
> - Domain-specific competitive DTW based STD system proposed by Anguera et al, 2013 where they perform subsequence-DTW search over Gaussian posterior features. In our implementation, we perform the search over posterior features extracted by Wav2Vec 2.0 ASRs.
>
> The results are added to Table 4 in the revised version of our manuscript.
> The results are discussed in the last $5$ lines of section $7.2$, Spoken Term Detection paragraph.
>
> Wav2Tok outperforms all the baselines.
>
>      6. No citations... claims.
>
>
>
> Response:
>
> We have added the necessary citations in  Section $1$ \textbf{Introduction} in the revised version of our manuscript for the convenience of the readers.
>
>
>
>      7. Are the MIP ... clarified.
>
>
> Response:
>
> No we didn't use NNFP for the experiments. We have clearly stated that we use their training approaches to train the same encoder network used in Wav2Tok in section $6.3$ baselines, MIPS and Triplet paragraphs.
>
>
>
>       8. While page 3 ... NNFP may be well suited.
>
>
> Response:
>
> Yes, we agree. We did not say that NNFP is not well suited for the task at hand.
>
>
>      9. Please use \citep ... say...
>
>
>
> Response:
>
>
>  We have edited and replaced all \cite with \citep in the revised version of the manuscript.
>
>        10. Data augmentations ... paper.
>
>
>
> Response:
>
> We did state which augmentations we apply in sub-section $7.1$ of our manuscript (the version which was reviewed).
>
>
>       11. Approximate  string matching  ..... citation .. provided.
>
>
> Response:
>
> We have added the citation in sub-section $6.2$  Spoken Term detection, Experiment Details paragraph in the revised version of the manuscript.
> We have also added a line citing the library which we use to perform approximate string matching, fuzzysearch.
> It automatically chooses the fastest algorithm for matching.
>
>
>     12. The description of wav2vec  baselines... stated.
>
>
>
> Response:
>
>
> Thanks for this comment.
> We have done the following revisions for more clarity,
>
> - We have rephrased the Wav2Vec 2.0 loss function in section $6.3$ Baselines, Wav2Vec2 paragraph.
> - We have rephrased section $6.3$ Baselines,  Wav2Vec2P paragraph to clearly state the difference between Wav2Vec2P and Wav2Vec2.
>
>
>
>
>
>
>
>     13. The dataset ... TIMIT? How ...introduction?
>
>
> Response:
>
> Thanks for the comment.
> We have added the details in section $6.2$ Experiment details lines 2,3,4 in  revised version of our manuscript.
>
>
>       14. Wav2Tok+Cos ...table 2
>
>
> Response:
>
> We apologize for the typing error. Wav2Tok+Cos has been presented in Table 2 as Wav2Tok+NoRel  in our manuscript (the version which was reviewed). We have resolved it in the revised version of the manuscript.
>
>
>      15. "Locally... hashing"
>
>
>
> Response:
>
> We apologize for the typing error. We have resolved it in the revised version of the manuscript.

---

> > ### Author Response · Authors · 2022-11-18
> > **Link to anonymous repository for Reproducibility (Codes)**
> >
> > We have added a link to an anonymous repository where we are uploading the codes. It is added as Supplementary material.

---

### Official Review · Reviewer_PPxN · 2022-10-27

**Confidence:** 4
**Clarity, Quality, Novelty And Reproducibility:** 1. Slightly difficult read.
2. I am …
**Correctness:** 2
**Technical Novelty And Significance:** 2
**Empirical Novelty And Significance:** 1
**Recommendation:** 8

**Strength And Weaknesses:**

Strengths:
1. The paper is easy to understand and the proposed approach seems to be interesting

Weakness
1. There is no clear explanation of why different aspects of the proposed approach are necessary.
2. The ablation studies presented are not good enough to understand the proposed approach thoroughly.
3. The authors test the proposed work on a few smaller datasets and claim their approach is better than popular approaches trained on large datasets. This conclusions seems a bit premature as the authors have not tested their approach on large datasets.
4. The paper is not really an easy read and the presentation can be improved.

**Summary Of The Paper:**

The authors address the problem of tokenizing audio data for improving the audio based content retrieval. They propose yet another approach for audio tokenization based on clustering the representation in Vector-Quantized space and forcing the representations to get closer their nearest centroid. The authors evaluate on smaller music and speech datasets for the query by humming and spoken term detection tasks and show better results compared to some of the existing approaches.

**Summary Of The Review:**

Overall, I feel this is yet another solution proposed for audio tokenization without clear articulation of what problems in the previous approaches are being solved. While this in itself is fine, the paper also does not contain any thorough justification for this design and seems like author's may have a good intuition about this problem which may have gone into the design but as a reviewer some aspects of the design are not at all clear.

---

> ### Author Response · Authors · 2022-11-17
> **Our approach gives concise discrete representations. Our training is performed in an EM algorithm fashion.  We have added another ablation study.**
>
> We thank the reviewer for the insightful comments. We address each of the questions and concerns as follows,
>
>       1. There is no clear explanation of why different aspects of the proposed approach are necessary.
>
>
>
> Response:
>
> Our goal is to find concise discrete representations of audio that can help in efficient retrieval. Orthography does it for an given language, but we aim at making an ML system do it for any audio sequences. In this paper, we learn these representations from pairs of similar audio without any other transcription (orthography) given.
>
> We use,
>
>
>    - K-means clustering: to find discrete representations (similar to E step of EM algorithm)
>
> -Contrastive loss: to explicitly map the continuous prototypes to the centroids (similar to M step of EM algorithm)
>
> -CTC loss: to reduce the edit distance between the discrete representation of the two audio in the given pair (similar to M step of EM algorithm)
>
>
> We have added this analogy with EM algorithm in section $5$ $4^{th}$ paragraph in the revised version of our manuscript.
>
> We have also added this to the abstract section in the revised version of the manuscript.
>
>
>      2. The ablation studies presented are not good enough to understand the proposed approach thoroughly.
>
>
> Response:
>
> We have presented the results of the following ablation studies
>
> - Wav2Tok+NoSim: remove the CTC loss
>
> -Wav2Tok+Cos : replace the parametric similarity in contrastive loss with simple cosine similarity
>
> Note Wav2Tok+Cos was presented as Wav2Tok+NoRel in table 2 in the version of the manuscript which was reviewed. We have resolved it in the revised version of the manuscript.
>
> We also added the following result to the revised version of our manuscript,
>
> - Wav2Tok+CTC: train with the CTC loss only
>
>
>
>
>
>       3. The authors test the proposed work on a few smaller datasets and claim their approach is better than popular approaches trained on large datasets. This conclusions seems a bit premature as the authors have not tested their approach on large datasets.
>
>
> Response:
>
> Thanks for this comment.
> To show the scalability of the proposed method, we have included experiments with Librispeech (for STD) and MIR1K (for QbH) tasks in the revised manuscript.
>
>     4. Overall, I feel this is yet another solution proposed for audio tokenization without clear articulation of what problems in the previous approaches are being solved. While this in itself is fine, the paper also does not contain any thorough justification for this design and seems like author's may have a good intuition about this problem which may have gone into the design but as a reviewer some aspects of the design are not at all clear.
>
>
> Response:
>
> No previous approach gave a concise discrete representation. While most other approaches give continuous representations, Wav2Vec2.0 gives discrete representations but even they are not concise and have not been used for any downstream task to the best of our knowledge.

---

> > ### Author Response · Authors · 2022-11-18
> > **Link to anonymous repository for Reproducibility (Codes)**
> >
> > We have added a link to an anonymous repository where we are uploading the codes. It is added as Supplementary material.

---

> > > ### Comment · Reviewer_PPxN · 2022-12-07
> > > **Thank you for a Detailed Response to the Reviewer Comments**
> > >
> > > I think the authors have done a pretty good job in addressing all the reviewer's concerns. I am also happy with the repsonse to the my comments. I am happy to change my score of the paper.

---

### Official Review · Reviewer_7rFd · 2022-10-28

**Confidence:** 4
**Correctness:** 2
**Technical Novelty And Significance:** 3
**Empirical Novelty And Significance:** 3
**Recommendation:** 3

**Clarity, Quality, Novelty And Reproducibility:**

**Clarity**

This paper is fairly clear overall, however the descriptions of key aspects of the proposed pre-training approach are quite vague. Specifically, the CTC-based alignment loss seems to be of critical importance to the success of this approach, however the explanation of this aspect in Section 5 leaves many questions unanswered. For instance, how is this loss function differentiated if it uses the discrete token sequence \tilde{T}’? Are there extra parameters that are trained to map the embedding vocabulary from Z to alphabet A U {CTC blank token}?

Another aspect which is unclear in this paper is how the training data is used in the query-by-humming example. Is the model pre-trained both on humming recordings and the full music recordings? Also, it seems a bit unconventional to train this representation on such a small / domain-specific dataset. Why not pre-train it on a large corpus of music recordings and then evaluate it on MIR-QbSH?

**Quality**

Perhaps the biggest issue with this work in its current form is the evaluation. The introduction of the paper appeals to the convenience of one-size-fits-all approaches over domain-specific strategies - as such, I expect to see a comparison where the proposed approach comes close to (or possibly even exceeds) the performance of domain-specific approaches.

Table 1 contains the “Triplet” baseline which appears to be a domain-specific approach, but it’s not stated whether this is representative of state-of-the-art for domain-specific query-by-humming. Additionally, the primary comparison seems to be against ED-based search with Wav2Vec2 tokens, but this is a straw man argument as Wav2Vec2 was not designed with ED-based search in mind (unlike Wav2Tok).

**Novelty**: The methods proposed here are reasonably novel - I am unaware of any other work which seeks to learn variable-rate discrete representations of audio that are useful for retrieval via text-based search.

**Reproducibility**: There is likely not enough information in this paper to reproduce the results, however the authors have promised to share code.

**Strength And Weaknesses:**

A strength of this paper is that it focuses on learning discrete representations of audio that are useful for edit distance-based text retrieval, in contrast to other work on discrete audio representations which does not consider text retrieval as a primary goal.

The primary weaknesses of this paper are (1) a lackluster evaluation which makes straw man comparisons against weak baselines and omits comparisons to domain-specific methods, and (2) an unclear description of critical aspects of the proposed methodology.

**Summary Of The Paper:**

This paper presents a discrete representation learning strategy for audio sequences designed primarily for retrieval. The learning strategy encourages representations to be similar for pairs of similar inputs, where pairs are created by data augmentation of the input audio. The authors compare their approach on two retrieval tasks: musical query-by-humming and spoken term detection.

**Summary Of The Review:**

Overall, I think this paper has some interesting ideas for a novel application of discrete representation learning for audio. However, a weak evaluation prevents me from understanding the strength of the proposed approach. I think the authors should continue working on this, but I don’t think it’s ready for ICLR at this time.

---

> ### Author Response · Authors · 2022-11-17
> **We have added domain-specific baselines for QbH and STD tasks. We have trained on larger dataset of Music recordings. We have also added  details of the CTC loss calculation**
>
>
> We thank the reviewer for the insightful comments. We address each of the questions and concerns as follows,
>
>
>     1. -  a lackluster evaluation which makes straw man comparisons against weak baselines and omits comparisons to domain-specific methods
>
>     - Perhaps the biggest issue ...  performance of domain-specific approaches.
>
>
>
> Response:
> Thanks for this comment.
>
> There are no existing works on the theoretical problem we mention, namely finding concise discrete representations for audio. The two applications of our work, namely spoken term detection and QbH have existing methods, which we have included (based on this comment) as baselines in the revised manuscript.
>
> The closest works to ours are as follows,
>
>
> - Wav2Vec 2.0 learns discrete representations from unlabelled audio, that are not concise and have not been used for any downstream task as far as we know.
>
> - MIPS, a contrastive learning approach to learn continuous representations from audio
>
> - Triplet, a well known learning approach to learning continuous representations from audio.
>
>
> We have compared our approach with all the above works.
>
>
> Based on the comment we have added the following baselines,
>
> QbH :
>
>
> We compare Wav2Tok with following domain-specific QbH baselines (cf Table 1),
>
> - a SOTA melody extraction algorithm by Salamon et al, 2012. which maps audio to midi.
>
> - a SOTA query by humming (QbH) system proposed by Mostafa and Fung, 2017
>
> The results are added to Table 1 in the revised manuscript.
> The results are discussed in the last $7$ lines of section $7.1$, Quality of Tokenization paragraph.
>
>
>
> STD :
>
>
> DTW based STD is a highly competitive baseline.
> We implement domain-specific STD system proposed by Anguera et al, 2013 where they perform subsequence-DTW search over Gaussian posterior features. In our implementation, we perform the search over posterior features extracted by Wav2Vec 2.0 ASRs.
> Wav2Tok outperforms such STD systems as well (cf. Table 4 in revised manuscript).
>
> The results are added to Table 4 in the revised manuscript.
> The results are discussed in the last $5$ lines of section $7.2$, Spoken Term Detection paragraph.
>
> Many other existing STD methods assume word boundaries to be given and cannot be used to find words in running speech. So we don't compare with them.
>
> The best baselines are presented in the main body of the revised version of our paper while others are presented in  Appendix.
>
>
> Note, the evaluation were carried out on unseen melody sequences (QbH) and OOV words (STD).
>
>
>
>     2.  This paper is fairly clear overall, however ... alphabet A U \{CTC blank token\}?
>
>
>
> Response:
>
> We thank the reviewer for pointing this out to us. We have now presented the CTC loss calculation and the loss itself in it's entirety.
> We  have also added  details on the CTC loss calculation in the revised version of our manuscript.
>
> The original CTC was proposed for ASR problem with text labels for supervision and used blanks. We propose CTC framework with no labels but only a pair of similar audio, and thus do not use blanks.
>
>
>
> We use cosine similarity scores with the codebook discrete representations to map embedding vocabulary from Z to alphabet A. Hence, no extra paramters are used for the mapping.
>
>
>
> Note, based on the review by reviewer cgKi,
>
>
> -we rephrased section $5$ Training paragraph $7$ Alignment loss as Likelihood Loss in the revised version of our manuscript.
> - We also rephrased all sentences corresponding to the CTC loss to omit the use of 'align'.
>
>
>  We request the reviewer to go through response $1$ in our official response to the review by reviewer cgKi.
>
>
>
>
>
>
>
>
>
>
>
>     3. Another aspect which is unclear ... a small / domain-specific dataset. Why.... large corpus of music recordings and then evaluate it on MIR-QbSH?
>
>
> Response:
>
> Thanks for this comment.
>
> Based on this comment, we also trained Wav2Tok on MIR-1K dataset which is composed of $1000$ full music recordings and evaluated performance in MIR-QbSH.
> We have added the results to Table $2$ in the revised version of our manuscript.
> It gives comparable performance to the domain-specific QbH baselines (cf. Table 1 in revised manuscript). The representations generated by Wav2Tok+MIR1K outperform  the baselines in search performance as well (see Appendix A.4).
>
> The results are discussed in lines $7$-$9$ in section $7.1$, Some Variations paragraph.
>
>
>
>
>
>
>
>
>
>
>
>
>
>
>     4. Table 1 contains the “Triplet” baseline .... Wav2Vec2  .... ED-based search in mind (unlike Wav2Tok).
>
>
>
> Response:
>
> Triplet and MIPS are representative of domain-specific query-by-example music approaches. We test these approaches on query-by-humming task.
>
> We have added two domain-specific QbH baselines in our paper which are outperformed by proposed Wav2Tok.
>
> We present a comparison of Wav2Tok representations with the representations generated by Wav2Vec 2.0 models in Appendix A.4.
> We observe Wav2Tok to generate better representations for music than the Wav2Vec 2.0 baselines.

---

> > ### Author Response · Authors · 2022-11-18
> > **Link to anonymous repository for Reproducibility (Codes)**
> >
> > We have added a link to an anonymous repository where we are uploading the codes. It is added as Supplementary material.

---

> ### Author Response · Authors · 2022-12-11
> **Request to review the rebuttal**
>
> Dear reviewer 7rFd,
> We have submitted our response to your concerns, including new experiments and the source code.
> Please let us know your comments.
> Very many thanks.

---

### Official Review · Reviewer_cgKi · 2022-10-31

**Confidence:** 3
**Correctness:** 3
**Technical Novelty And Significance:** 3
**Empirical Novelty And Significance:** 3
**Recommendation:** 8

**Clarity, Quality, Novelty And Reproducibility:**

The use of English is good.  There are only a few typo errors (e.g. "of of").

The paper comprises multiple complicated ideas, and the authors do an adequate job of explaining them within the constraints of the page limit.

The description of the approach, together with some educated guessing, should allow for replication of the experimental procedure.  However, the modified datasets that were synthesised from existing public datasets have not been released by the authors yet.


**Strength And Weaknesses:**

Strengths:
- The problem that the proposal in this paper is trying to address is well motivated.
- The proposal is tested on multiple diverse tasks.
- The proposal is explicitly compared against multiple baselines.

Weaknesses:
- In the paper, it is emphasised that the CTC loss is used for the purpose of aligning two audio sequences.  The use of the word "align" in this context may lead to misinterpretation, as it differs from the standard use of the word.  The standard meaning of "align" is to determine which input audio frames correspond to each token in the output.  This is not what CTC is being used for in this paper.  If this is what CTC is being used for in this paper, then this use comes into conflict with the BLSTM topology of the model, because prior works have shown that the temporal flexibility of a BLSTM allows it to learn any arbitrary alignment between the input and output, without needing any realistic correspondence.  The actual use of the CTC loss in this paper is to encourage the model to output similar token sequences for both inputs, which is not what "align" typically means.  Perhaps the authors may consider re-phrasing the description of the motivation for the CTC loss, to abate potential confusion.
- Wav2Tok+Cos is described in the text, and the text seems to imply that the performance results can be found somewhere.  However, these results seem to be missing from table 2.
- During audio retrieval, the query is compared against the database using token sequence edit distances.  This assumes that all non-similar tokens are equidistant from each other.  This equidistance assumption may potentially limit the system performance.  This assumption is not discussed in the paper.
- The unit of measurement in many of the experiment tables is not stated in the table, and a reader would need to search through the text to figure out what the numbers in the tables represent.


**Summary Of The Paper:**

This paper proposes an unsupervised method of computing discrete token sequences from audio, for the purpose of audio retrieval.
Previous works either:
- Use supervised token sequence labels during training, which requires domain expertise and access to labelled data.
- Train an unsupervised autoencoder on only a single sequence at a time, and are therefore not contrastive in nature.

The proposal in this paper addresses both of these issues, by contrastively discovering a token codebook. A local minimum, where all codebook entries converge to the same value, is prevented by the joint use of a softmax contrastive criterion and a regular K-means re-clustering.


**Summary Of The Review:**

The proposed approach seems sufficiently novel.  The proposal seems sufficiently useful in addressing limitations of previous methods.  I do not find any major flaws in the paper.  The experiments seem sufficiently comprehensive.

The Ithenticate similarity is 3%, which is good.

---

> ### Author Response · Authors · 2022-11-17
> **We have rephrased the CTC loss details and mentioned the equidistance assumption**
>
>
> We thank the reviewer for the insightful comments. We address each of the questions and concerns as follows,
>
>     1. In the paper, it is emphasised that the CTC loss is used for the purpose of aligning two audio sequences. The use of the word "align" in this context may lead to misinterpretation, as it differs from the standard use of the word. The standard meaning of "align" is to determine which input audio frames correspond to each token in the output. This is not what CTC is being used for in this paper. If this is what CTC is being used for in this paper, then this use comes into conflict with the BLSTM topology of the model, because prior works have shown that the temporal flexibility of a BLSTM allows it to learn any arbitrary alignment between the input and output, without needing any realistic correspondence. The actual use of the CTC loss in this paper is to encourage the model to output similar token sequences for both inputs, which is not what "align" typically means. Perhaps the authors may consider re-phrasing the description of the motivation for the CTC loss, to abate potential confusion.
>
>
> Response:
>
>  We thank the reviewer for this comment.
>
> Based on this comment,
>
>
>
>    - We have rephrased section $5$ paragraph $7$ Alignment Loss as Likelihood loss in the revised version of our manuscript.
>
>  - We  rephrased section $5$ paragraph $3$ last line.
>
>  - We rephrased page $2$ paragraph $3$ last line as minimizing the edit distance between token sequences.
>
>
>
>
>
>
>
> In addition, we added the details to our CTC loss calculation in section $5$ paragraph $7$ Likelihood loss.
> Note, the CTC loss function has been presented in its entirety based on the reviews given by reviewer U7Da and 7rFd.
>
>
>
>
>
>
>
>
>      2. Wav2Tok+Cos is described in the text, and the text seems to imply that the performance results can be found somewhere. However, these results seem to be missing from table 2.
>
>
>
> Response:
>
>  We apologize for the typing error. Wav2Tok+Cos was presented in Table 2 as Wav2Tok+NoRel in our manuscript (the version which was reviewed). We have corrected it in the revised version of the manuscript.
>
>
>
>     3. During audio retrieval, the query is compared against the database using token sequence edit distances. This assumes that all non-similar tokens are equidistant from each other. This equidistance assumption may potentially limit the system performance. This assumption is not discussed in the paper.
>
>
> Response:
>
> Thanks for this suggestion. We have indeed used the equidistant assumption. We have included this in section $8$ Conclusion and Future Work in the revised version of the manuscript.
>
>
>
>     4. The unit of measurement in many of the experiment tables is not stated in the table, and a reader would need to search through the text to figure out what the numbers in the tables represent.
>
>
> Response:
>
>
> Thanks for pointing it out.
> We have added the unit of measurement (MRR) in the respective columns of Table 3 in the revised version of the manuscript.
>
> We did mention the unit of measurement in the caption of Table 1,2 ('MRR Scores for query by humming')in our manuscript (the version which was reviewed). In Table 4: Spoken Term Detection, we have added the unit of measurement '(F1-scores)'  to the caption.

---

> > ### Author Response · Authors · 2022-11-18
> > **Link to anonymous repository for Reproducibility (Codes)**
> >
> > We have added a link to an anonymous repository where we are uploading the codes. It is added as Supplementary material.

---

### Official Review · Reviewer_U7Da · 2022-11-03

**Confidence:** 3
**Correctness:** 4
**Technical Novelty And Significance:** 4
**Empirical Novelty And Significance:** 3
**Recommendation:** 8

**Clarity, Quality, Novelty And Reproducibility:**

The paper is written mostly clearly. The results seem reproducible. It would help to provide the software.

**Details Of Ethics Concerns:**

No ethics concerns.

**Strength And Weaknesses:**

Strengths:
1. A novel method to extract token sequences from audio signals, trainable with pairs of audio signals that should correspond to each other.
2. Comparison of the method with competitive methods on two different audio retrieval tasks.

Weaknesses:
1. Uses BiLSTM models and shows that it is better than using a transformer architecture. For audio, U-net style models are also used in token generation which could be compared.
2. Similarity and difference between the first part of the loss and VQ-VAE loss would be a nice addition to the paper.
3. Details about how the CTC loss is back-propagated to the encoders is not clear. Does the CTC model use blanks? What happens when the label sequence is longer than the number of frames?

**Summary Of The Paper:**

The paper describes a deep neural network method for discrete tokenization of audio signals for retrieval problems. The model is trained from pairs of audio that correspond to each other. The tokenization has two losses, one is similar to a contrastive version of a VQ-VAE loss and the other is a loss that encourages correspondence of representation between two pairs of audio. This second loss uses CTC which correlates with a low edit distance between pair token sequences.

The method is applied to query by humming and spoken term detection tasks and it is shown to perform better than other alternatives including using large self-supervised representations (wav2vec 2.0) etc.

**Summary Of The Review:**

The method seems novel and seems worth publishing.

Some specific comments are given below:

1. Is the contrastive loss in (3) back-propagated to both p and e?  In VQ-VAE, it is not done to avoid collapse. But here, due to the contrastive nature of the loss, we do not have to worry about that? Some explanations would be nice.
2. In Table 1, what are the columns V, TS and PS? Validation, test and something else?
3. In Table 2, there is no mention of Wav2Tok+Cos which is mentioned in the text. Is it equivalent to Wav2Tok+NoRel in the table which is not mentioned in the text.
4. Do the losses apply to both of the inputs in a pair? Maybe this can be stated more clearly.
5. If the token sequence is longer than the number of frames in the other audio, how does the CTC loss work? Typically, CTC loss assumes the label sequence is shorter than the number of frames and uses blank symbols to fill in the middle.

---

> ### Author Response · Authors · 2022-11-17
> **We have added the details to CTC loss calculation and loss (3) backpropagation**
>
>
> We thank the reviewer for the insightful comments. We address each of the questions and concerns as follows,
>
>
>    1. Uses BiLSTM models .... transformer architecture..... U-net... compared.
>
>
> Response:
>
> The experiments highlight the advantage of the proposed training method over those used in the baseline approaches.
> We do not compare model architectures or types.
> The same method can be adopted to other architectures as well.
>
>
>
>    2. Similarity and difference.. ... VQ-VAE loss... paper.
>
>
> Response:
>
> The VQ-VAE loss has three terms ,
>
>    - reconstruction loss , VAE loss but with discrete representations
>    -  L-2 loss pushing the discrete representations close to corresponding encoder representations with stop gradient applied to encoder.
>    - L-2 loss pushing the encoder representations close to corresponding discrete representations with stop gradient applied to discrete representation. This loss has a much lesser weightage.
>
>
> The  two L-2 losses together form a $K$-means loss. Such splitting of the loss function ,adding lesser weightage to the second L-2 loss and usage of stop gradient avoids collapse.
> The codebook and model parameters are learnt jointly in one pass.
>
> In  our work, loss (3) doesn't update the discrete representations in codeboook. We use an offline clustering step to update the codebook to avoid collapse.
>
>
>     3. Details about how the CTC loss ... frames?
>
>
> Response:
>
> We have addressed this issue by adding more details about our CTC training .
>
> The original CTC was proposed for ASR problem with text labels for supervision and used blanks. We propose CTC framework with no labels but only a pair of similar audio, and thus do not use blanks.
>
>
> We rephrased the CTC loss paragraph accordingly and have added  details on the CTC loss calculation in the revised version of our manuscript.
>
> Note, based on the review by reviewer cgKi,
>
>
>      -  we rephrased section $5$ Training paragraph $7$ Alignment loss as Likelihood Loss in the revised version of our manuscript.
>
>      -  We also rephrased all sentences corresponding to the CTC loss to omit the use of 'align'.
>
>
>  We request the reviewer to go through response 1 in our official response to the review by reviewer cgKi.
>
>
>
>
>
>
>
>
>
>     4. Is the contrastive loss ..... be nice.
>
>
> Response: The contrastive loss $\mathcal{L}_m$ (loss (3)) is backpropagated to p only and not e to avoid collapse.
> The codebook discrete representations e are updated iteratively via offline clustering.
>
> In the revised version of our manuscript we have cleared this confusion by adding a stop gradient operator on e in $\mathcal{L}_m$ (loss (3)).
>
> In addition,
>
>     - We have added a line to $2^{nd}$ paragraph of section $5$ Training to present how the discrete representations are initialised.
>
>     -  We have also added  $3$ lines in $4^{th}$ paragraph of section $5$ Training to present that the training is performed in a manner similar to Expectation Maximization algorithm. The clustering is used as the E-step and the backpropagation of $\mathcal{L}$ (loss (1)) as the M-step.
>
>
>
>
>
>
>
>
>
>    5. In Table 1, what are the columns V, TS and PS? Validation, test and something else?
>
>
> Response:
>
>     - V- Vanilla Query, results for queries  with no augmentation
>     -  TS- Time Stretched query, results for the same queries with time stretch augmentation
>     - PS - Pitch shifted query , results for the same  queries with pitch shift augmentation
>
>
> We have mentioned what  V, PS, TS stand for in first paragraph of Section 7.1 in our manuscript (the version which was reviewed).
>
>
>
>     6. In Table 2, there is no mention of Wav2Tok+Cos ......Wav2Tok+NoRel ..... in the text.
>
>
> Response: We apologize for the typing error. Wav2Tok+Cos has been presented in Table 2 as Wav2Tok+NoRel  in our manuscript (the version which was reviewed). We have resolved it in the revised version of the manuscript.
>
>
>    7. Do the losses apply to both of the inputs in a pair? Maybe this can be stated more clearly.
>
>
> Response: Yes the losses apply to both the sequences.
> We have presented our overall loss $\mathcal{L}$ (loss (1)) in it's entirety in the revised version of our manuscript to abate such confusion.
>
>
>
>
>
>
>
>    8. If the token sequence is longer than the number of frames in the other audio, how does the CTC loss work? Typically, CTC loss assumes the label sequence is shorter than the number of frames and uses blank symbols to fill in the middle.
>
>
> Response: We do not calculate the loss for such cases.
>
> Note, in our total loss $\mathcal{L}$ (loss (1) in the revised version of our manuscript) we simply put $\alpha = 0$ if sequence $\mathcal{X}$ is smaller than sequence $\tilde{\mathcal{T}'}$ and $\beta = 0$ if sequence $\mathcal{X}'$ is smaller than sequence $\tilde{\mathcal{T}}$.

---

> > ### Author Response · Authors · 2022-11-18
> > **Link to anonymous repository for Reproducibility (Codes)**
> >
> > We have added a link to an anonymous repository where we are uploading the codes. It is added as Supplementary material.

---

> > ### Comment · Reviewer_U7Da · 2022-12-02
> > **Post-rebuttal note from reviewer**
> >
> > Thanks for the response to my review.
> >
> > I think the paper got better and I increased my score to 8.
> >
> > Your VQ-VAE update is similar to exponential moving average (ema) updates for the codebook, but you use an offline clustering step instead. You may try ema updates to the codebook too.
> >
> > In the original CTC, the labels are not repeated but blanks are inserted in between labels. I guess in your version, you do not use blanks but allow repetition of the labels to determine different paths through the compressed label sequence. it would help to mention this in the paper.

---

### Author Response · Authors · 2022-11-18
**Thank you Reviewers for the insightful Comments and Reviews !!**

We thank the reviewers for their insightful comments and reviews. We have addressed each of the questions/ concerns the reviewers have raised.
We have also added a link to the anonymous repository where we are uploading the codes. The link is added as supplementary material.



A brief summary of the revisions made based on the reviews by the reviewers,


Clarity and Presentation:




- We have presented the loss functions in their entirety

- We have added the details of our CTC loss calculation with further details on the CTC forward-backward variables in Appendix B

- We have rephrased $\mathcal{L}_{ctc}$ loss (4) as likelihood loss and all sentences containing the word 'align' as 'maximising the likelihood'

- We have presented (addition of 2 line to Section 5 Training paragraph 4) a clearer view of our algorithm (EM algorithm like training, E-step - Clustering M-step - Backpropagation of $\mathcal{L}$ loss (1))

- We have cleared out the confusion on the backpropagation of $\mathcal{L}_m$ (loss (3)).

- We have added Clearer presentations of the training approaches of baselines Wav2Vec2  and Wav2Vec2P

- We have rephrased parts of the abstract for  more clarity

Experiments:




- We have included 2 SOTA domain specific QbH baselines and 2 highly competitive domain specific STD baselines

- We add log-mel features and tokens obtained via quantization of log-mel features as baselines.

- We have included results of our model when trained on larger datasets like LibriSpeech-100 hours (for STD) and MIR-1K (for QbH).
Wav2Tok trained on LibriSpeech is evaluated for STD in TIMIT.
Wav2Tok trained on MIR-1K polyphonic music recordings is evaluated for QbH on MIR-QbSH monophonic humming recordings.

- We add another ablation study where we train Wav2Tok with our pairwise CTC loss only.

- We have added full version of table 2 to Appendix A.3

- We have added the performance of the representations generated by the models in Appendix A.4

---

### Decision · Program_Chairs · 2023-01-20

**Decision:**

Accept: poster

**Justification For Why Not Higher Score:**

There are still some issues with the work, although they are considered relatively minor.

1. The proposed Wav2Tok is supposed to be robust (through quantization) and fast (through shortened tokenization sequences leveraging CTC).  Although the speed advantages have been demonstrated in the experiments, the robustness is only partially verified.

2. As a generic technique that can be used for QbH and STD,  how does it compare with domain-specific techniques (e.g.  OOV issues in STD, generalization capability to other domains, etc..) is still not very clear.


**Justification For Why Not Lower Score:**

The idea is interesting.  The work is technically novel.  The experimental results are good.

**Metareview: Summary, Strengths And Weaknesses:**

In this paper the authors propose a robust and fast audio retrieval approach that is based on concise discrete representation of an audio sequence.  The audio retreival problem is then converted to a matching problem between the tokenized query and target sequences.  The learning of the discrete representations is conducted using the paired signals of the original audio and its data augmented version based on CTC and quantization.  Experiments on query by humming (QbH) and spoken term detection (STD) tasks show supportive results.   The authors cleared most of the concerns raised by the reviewers in their rebuttal, especially on the motivation, justification and implementation details of CTC which plays a crucial role in the learning.  The authors also added new experiments to verify the performance of Wav2Tok on additional datasets and domain-specific baselines.  All reviewers consider the idea interesting and the work technically novel.  The performance is good on the two investigated tasks although there are lingering minor issues that need to be taken care of in the final version.  Last but not least, the paper has a formatting issue where the fonts are too small in the tables.

**Note From Pc:**

if the above contains the word "oral" or "spotlight" please see: "oral" presentation means -> notable-top-5% and "spotlight" means -> notable-top-25%. As stated in our emails, we are disassociating presentation type from AC recommendations

**Summary Of Ac-Reviewer Meeting:**

A virtual meeting took place between AC and 4 of the 5 reviewers.

1.  All reviewers consider the idea interesting and the work is technically novel.

2.  The proposed Wav2Tok is supposed to be robust (through quantization) and fast (through shortened tokenization sequences leveraging CTC).  Although the speed advantages have been demonstrated in the experiments, the robustness is only partially verified.

3. As a generic technique that can be used for QbH and STD,  how does it compare with domain-specific techniques (e.g.  OOV issues in STD, generalization capability to other domains, etc..) is still not very clear.

4. Small fonts in 4 tables in the main body of the paper.  This may result in a space violation if the fonts are the normal size.

The above lingering concerns are considered relatively minor and can be either fixed in the final version or further explored in the future work.  But these are not considered killers. Otherwise it would be too much to ask in one paper.

 One reviewer (reviewer 7rFd) did not get involved in the discussion and meeting. The AC and the reviewers went through its comments in the meeting and thought that most of his comments had been taken care of in the authors' rebuttal.